# Geographical and disciplinary coverage of open access journals: OpenAlex, Scopus, and WoS

**Abdelghani Maddi**[1]*, **Marion Maisonobe**[2], **Chérifa Boukacem-Zeghmouri**[3]

**1** Sorbonne Université, CNRS, Groupe d'Étude des Méthodes de l'Analyse Sociologique de la Sorbonne, GEMASS, F-75017 Paris, France, **2** Laboratoire Géographie-cités, CNRS, Université Paris 1, Université Paris Cité, EHESS, Aubervilliers, France, **3** Université Claude Bernard Lyon 1, Villeurbanne, France

* abdelghani.maddi@cnrs.fr

**Data availability statement:** The data are publicly available on Zenodo and can be

## Abstract

This study aims to compare the geographical and disciplinary coverage of OA journals in three databases: OpenAlex, Scopus and the Web of Science (WoS). We used the Directory of Open Access Scholarly Resources (ROAD), provided by the ISSN International Centre, as a reference to identify OA active journals (as of May 2024). Among the 62,701 active OA journals listed in ROAD, the WoS indexes 6,157 journals, Scopus indexes 7,351, while OpenAlex indexes 34,217. A striking observation is the presence of 24,976 OA journals exclusively in OpenAlex, whereas only 182 journals are exclusively present in the WoS and 373 in Scopus. The geographical analysis focuses on two levels: continents and countries. As for disciplinary comparison, we use the ten disciplinary levels of the ROAD database. Moreover, our findings reveal a similarity in OA journal coverage between the WoS and Scopus. However, while OpenAlex offers better inclusivity and indexing, it is not without biases. The WoS and Scopus predictably favor journals from Europe, North America and Oceania. Although OpenAlex presents a much more balanced indexing, certain regions and countries remain relatively underrepresented. Typically, Africa is proportionally as under-represented in OpenAlex as it is in Scopus, and some emerging countries are proportionally less represented in OpenAlex than in the WoS and Scopus. These results underscore a marked similarity in OA journal indexing between WoS and Scopus, while OpenAlex aligns more closely with the distribution observed in the ROAD database, although it also exhibits some representational biases.

## Introduction

The academic publishing landscape is experiencing a seismic shift as more researchers, institutions, and funding bodies embrace open access (OA) publishing models [1]. This transition is driven by a multifaceted set of factors, including policy mandates, institutional initiatives, changing attitudes towards scholarly communication, as well as economic and philosophical considerations that influence the support for and adoption of open access practices [2]. From an economic perspective, the rising costs of subscription-based models have prompted

accessed at the following link: https://zenodo.org/records/14389358.

**Funding:** The author(s) received no specific funding for this work.

**Competing interests:** The authors have declared that no competing interests exist.

universities and research institutions to seek more sustainable alternatives that allow wider access to research outputs without the financial burden. Open Access is based on the philosophical and democratic principles that rely on greater transparency, equity, and accessibility in scholarly publishing, with many arguing that publicly funded research should be freely accessible to all. OA goals seek to democratize knowledge, reduce barriers to information, and promote the free flow of research across disciplines and borders. Governments, research funders, and academic institutions worldwide are increasingly recognizing the benefits of making research outputs freely accessible to all, without barriers such as subscription fees or paywalls [3].

In response to these incentives, there has been a proliferation of OA journals and platforms, facilitated by various funding models [4,5]. One notable development is the emergence of the Diamond model, characterized by journals that are both OA for readers and free of publication charges for authors [6,7]. Traditional bibliographic databases like Web of Science (WoS) and Scopus were developed on the basis of the subscription-based journals, and their coverage did not fully take into account the development of the growing prevalence of OA publishing [8]. While these databases have made efforts to incorporate OA content, they often lag behind in terms of coverage and inclusivity [9]. This discrepancy is particularly evident when comparing their journal coverage to that of the Directory of OA Scholarly Resources (ROAD), which exclusively indexes OA journals. Indeed, the traditional databases do not take openness as a criterion among their indexing criteria. WoS and Scopus both have a list of indexing criteria including "quality" criteria (i.e., the presence of a peer review policy, article titles and article abstracts in English, timeliness, international Editorial Boards) and "impact" criteria, which somehow means that the published articles must receive citations from journals already indexed in these databases. As of December 2024, the WoS indexing criteria are summarized here: https://clarivate.com/academia-government/scientific-and-academic-research/research-discovery-and-referencing/web-of-science/web-of-science-core-collection/editorial-selection-process/journal-evaluation-process-selection-criteria/. Scopus indexing criteria (less numerous and precise than WoS criteria) are summarized here: https://www.elsevier.com/products/scopus/content/content-policy-and-selection. Because of this last dimension, the indexing closely depends on the current content of the databases, reinforcing their initial biases. By adhering to these restrictive standards, these databases fail to fully represent the breadth of open access publishing, which has grown substantially in recent years. As a result, the traditional databases are limited in their ability to reflect the diversity and evolution of scholarly publishing, leading to a skewed representation that overlooks a significant portion of the academic literature.

Enter OpenAlex, a new player in the field of bibliographic databases, heralded at least in France and some European countries, as a potential game-changer in the realm of OA publishing [10]. OpenAlex's agenda is to address the limitations of traditional databases by providing a more comprehensive and inclusive index of scholarly journals [11]. By leveraging advanced data harvesting techniques and partnerships with academic institutions, OA repositories, and publishers, OpenAlex aims to offer a broader and more diverse representation of scholarly outputs from around the world [12]. The current key question surrounding OpenAlex is whether it can effectively address the historical biases and limitations of traditional databases. While OpenAlex holds promise as a potential solution to these long-standing issues, its effectiveness has yet to be fully evaluated. Ongoing research and analysis will be essential to assess the impact of OpenAlex on the visibility, accessibility, and diversity of scholarly publications. As the academic publishing landscape continues to evolve, OpenAlex represents a significant development in the ongoing quest for a more open, inclusive,

and equitable scholarly communication ecosystem. Aiming to contribute to this topic, our research questions are:

RQ1: How do traditional and alternative scholarly databases differ in their representation of OA journals across macro regions and income groups?

RQ2: Does OpenAlex possess broader disciplinary representation than traditional databases (WoS and Scopus)?

RQ3: Can OpenAlex overcome the challenges of disciplinary biases and ensure a more equitable representation of research outputs across diverse fields?

In this study, our objective is to conduct a comparative analysis of the geographical and disciplinary coverage of OA journals across three prominent databases: the WoS, Scopus, and OpenAlex. Through this approach, we aim to shed light on the differences and nuances in OA journal coverage across the three databases. For this purpose, we employ the ROAD database as a coverage reference as it provides a comprehensive catalogue of 62,701 indexed OA journals. Our analysis focuses on two key dimensions: geographical representation and disciplinary diversity. Geographically, we examine coverage across continents, income groups and countries. Additionally, we conduct a disciplinary comparison using the ten disciplinary levels outlined in the ROAD database. The article is structured as follows: we start with a review of the literature on recent papers comparing the three databases. Then, we present the data collected and method used aligned on the study objectives. The results encompass an analysis of the overall coverage of OA journals and a comparison of their geographical and disciplinary structure, including an examination at the country, income group and continent levels. Finally, we provide a discussion of the findings and their implications.

## Literature review

While there is a growing number of studies using OpenAlex as a data source, few have focused specifically on OpenAlex's OA coverage and limitations discussed in this paper. This is why our literature review focuses solely on comparative studies involving OpenAlex alongside other sources, a choice aligned with our research questions.

A recent study by Alperin et al. [12] observed a growing trend in using OpenAlex as a data source. They compared OpenAlex and Scopus data, finding more publications in OpenAlex, particularly from regions and languages under-represented in Scopus. However, they identified areas for improvement in metadata accuracy (e.g., affiliations, document types, open access status) and completeness in OpenAlex. Another study by Culbert et al. [13] explored OpenAlex as a promising open source of scholarly metadata, comparing it with WoS and Scopus (whose metadata are not error-free either). They assessed reference and metadata coverage, demonstrating OpenAlex's comparability in reference numbers but mixed results in other metadata. Their study highlighted the importance of addressing data and metadata trustworthiness in rapidly evolving sources like OpenAlex.

Jiao et al. [14] investigated the indexing of data papers in scholarly databases to understand how research data is published and reused. They examined 18 data journals across WoS, Scopus, Dimensions, and OpenAlex to evaluate coverage and document type information consistency. Their findings revealed highly inconsistent coverage of data papers and their document types across databases, posing challenges for quantitative analysis. Jiao et al. [14] showed that, while newer databases like Dimensions and OpenAlex cover all data journals, they classify data papers as regular research articles, making their retrieval challenging. In contrast, although Scopus and the WoS cover fewer data journals, they distinguish data papers with a

'Data paper' document type. However, inconsistencies persist, indicating a need for improved communication to enhance database quality.

Ortega and Delgado-Quirós [15] analysed retractions and withdrawals in scholarly databases, highlighting differences between traditional citation indexes like the WoS and newer hybrid databases like OpenAlex. Their findings underscored the impact of database selection on coverage of retractions and withdrawals. Ortega and Delgado-Quirós [15] high-lighted that the differences primarily stem from how withdrawals are indexed by newer hybrid databases like Dimensions, OpenAlex, Scilit, and The Lens. Excluding withdrawal data, OpenAlex and The Lens collect the most retractions, while Scilit, Scopus, and Dimensions include the highest number of retracted articles. This suggests a distinction between traditional citation indexes such as WoS, PubMed, and Scopus, which are journal-based and do not index withdrawals, and newer hybrid databases relying on external sources like Crossref and Microsoft Academic. Since September 2023, Crossref, the leading DOI registration agency, acquired the RetractionWatch database, making it freely accessible [16]. Consequently, OpenAlex now directly incorporates RetractionWatch data to enrich its retraction field (see https://docs.openalex.org/api-entities/works/work-object). More recently, Delgado-Quirós and Ortega [17] aimed to compare metadata completeness across academic databases. They found that third-party databases like OpenAlex had higher metadata quality and completeness compared to academic search engines like Google Scholar. Their study emphasized the need for reliable descriptive data retrieval, especially in third-party databases.

Moreover, other studies pointed out that the metadata quality of OpenAlex has significant room for improvement to be usable in bibliometric studies. For example, Zhang et al. [18] investigated missing institutional information in journal article metadata in OpenAlex. They identified significant gaps, particularly in early years and social sciences and humanities disciplines. Their study emphasized the importance of data quality improvements in open resources like OpenAlex. Similarly, Bordignon [19] discussed the growing adoption of OpenAlex, citing institutions' decisions to transition from proprietary bibliometric products, notably the decision of Sorbonne Université in France to unsubscribe from WoS (see https://urls.fr/_8o4s2), and CNRS (French National Centre for Scientific Research) unsubscribing from Scopus (see https://urls.fr/PtYcj_) but keeping WoS subscription. She highlighted the importance of assessing the relevance of OpenAlex for bibliometric analysis, presenting tests to evaluate its effectiveness at an institutional level. Bordignon [19] came out with similar conclusions to Zhang et al. [18] regarding the quality of institutional metadata, based on a case study of publications from École des Ponts (a French engineering school). More recently, Céspedes et al. [20] assessed the linguistic coverage of OpenAlex and the accuracy of its metadata compared to the WoS. Through an in-depth manual validation of 6,836 articles, the study found that OpenAlex offers a more balanced representation of non-English languages than the WoS. However, the language metadata was not always accurate, leading to an overestimation of English and an underestimation of other languages. This research underlined the need for infrastructural improvements to ensure accurate metadata, despite OpenAlex's potential for comprehensive linguistic analysis in scholarly publishing.

Another study from Alonso-Alvarez and van Eck [21] examined the coverage and metadata availability of African publications in OpenAlex, comparing it with Scopus, WoS, and African Journals Online (AJOL). Their findings revealed that OpenAlex offers the most extensive coverage of African-based publications, but still lags in providing detailed metadata, particularly regarding affiliations, references, and funder information. Interestingly, metadata

completeness was found to be better for publications indexed in both OpenAlex and the proprietary databases, highlighting areas for improvement in OpenAlex to better serve research from the Global South.

## Data

Data collected for the study was primarily sourced from the Directory of OA Scholarly Resources (ROAD) (https://www.issn.org/services/online-services/road-the-directory-of-open-access-scholarly-resources/), kindly provided to us by the ISSN International Centre (https://www.issn.org/) in XML format. We extracted information for each of the 62,701 indexed sources (ISSN), including country (across 163 countries) and disciplines (10 levels). It is noteworthy that some journals lacked information regarding their discipline (3.6%—2,263 journals out of 62,701).

The ROAD database primarily indexes OA journals, but also includes other types of OA resources, such as monographic series, which are considered part of the broader open access publishing landscape. As shown in Fig 1, the major share of resources in ROAD are journals: 90.45% of the resources in ROAD are journals, representing 56,714 out of 62,701 entries. For consistency and clarity, the term 'journal' is used throughout the paper to refer to all

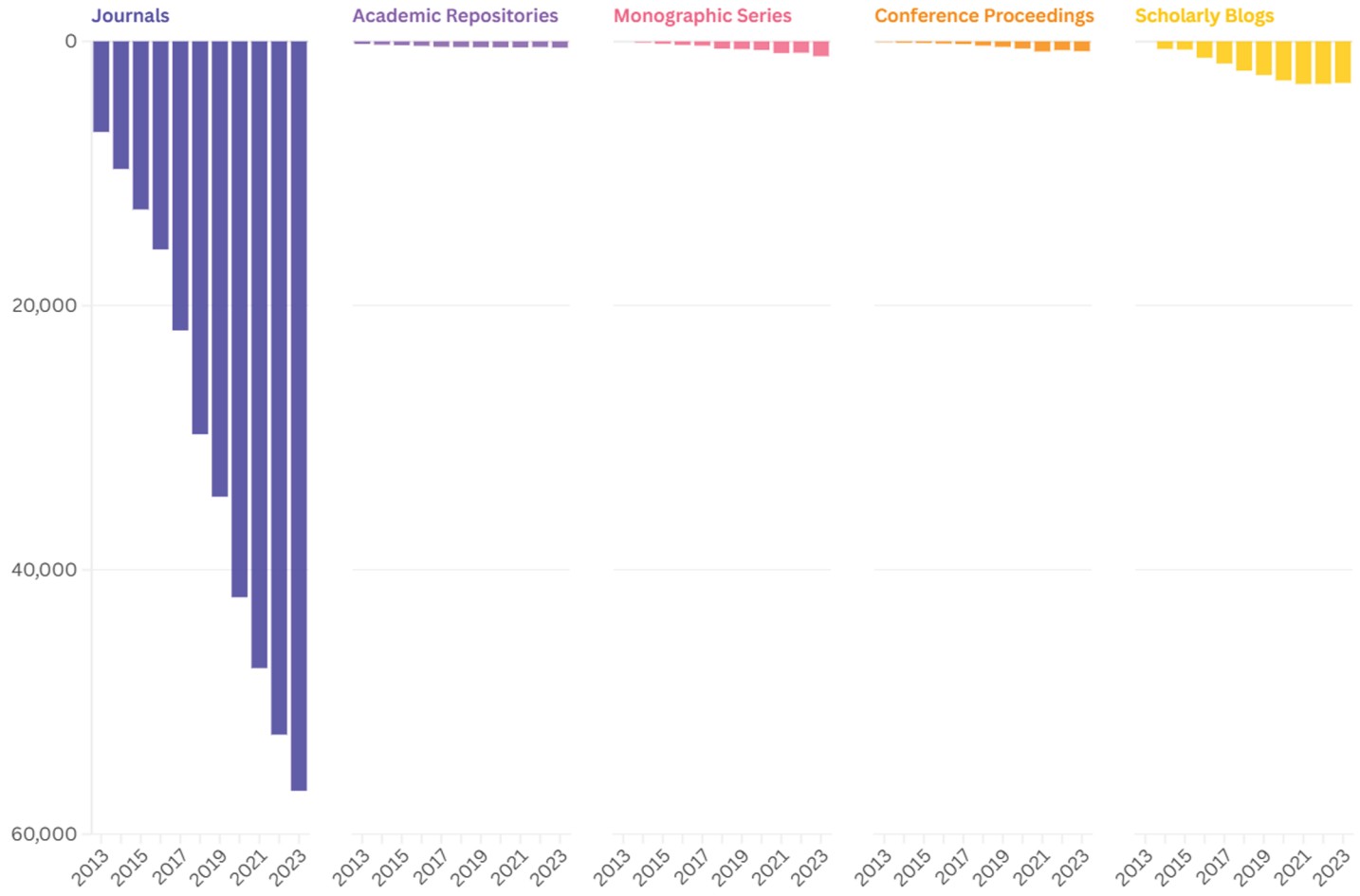

**Fig 1. Distribution of resources in ROAD database by year.**

types of resources indexed in ROAD, as they follow the same editorial standards and article publication processes.

Launched in late 2013, ROAD offers free access to a subset of bibliographic records from the ISSN Portal, describing scholarly resources available in OA identified by an ISSN/eISSN. These resources include Journals, Monographic series, Conference proceedings, Scholarly blogs, and Academic repositories. Metadata for these records, created by the ISSN network comprising 93 national centres and the ISSN International Centre, are enriched with data from indexing services, directories (such as DOAJ, Latindex, The Keepers Registry), and performance indicators (Scopus). ROAD is aligned with UNESCO's efforts to promote OA to scholarly resources and complements the Global Open Access Portal (GOAP) (https://www.goap.info/) developed by UNESCO, which provides an overview of OA to scholarly information worldwide.

ROAD applies specific inclusion criteria to list resources in its directory. To be included, a resource needs several requirements, including being freely accessible without registration, providing a clear description of its OA policy and licensing terms, and presenting scholarly content across various fields. The resource must also have clear editorial responsibility, academic affiliation, publishing entity, and adhere to ethical guidelines and indexing standards.

For the purpose of this study, ROAD was used as the gold standard for comparing the coverage of OpenAlex, Scopus, and WoS in OA journals. ROAD includes exclusively OA journals and provides comprehensive metadata for each journal, including ISSN, eISSN, country, and discipline. We included all journals from ROAD with complete metadata, without applying any additional filters. The data extraction from ROAD was conducted in October 2023, yielding 253,200 identifiers (ISSN and/or eISSN) for 183,158 distinct journals.

For our analysis, we focused on active OA journals in OpenAlex that had more than five publications. From the total of 183,158 journals in OpenAlex, 6,632 journals with no more than five publications were excluded from the analysis. Consequently, we considered 176,526 active journals from OpenAlex. Additionally, we included 29,262 "active" journals from Scopus and 23,189 'active' journals from the WoS Core Collection (covering AHCI, SSCI, SCIE, and ESCI). For these two databases (WoS and Scopus), we used the information they provide on the status of their indexed journals.

In addressing classification differences, we used the metadata provided by ROAD for the country and discipline classification, as ROAD's classifications are considered reliable, being provided by the ISSN Centre. This approach allowed us to bypass the issue of reconciling different classification systems between the databases. Our comparison therefore focused on the indexing of OA journals listed in ROAD across the other databases (OpenAlex, WoS, and Scopus).

It is important to note here that our methodology was specifically designed to avoid potential biases related to metadata inaccuracies in OpenAlex. By relying on ROAD as the gold standard, which provides comprehensive and reliable metadata for OA journals, we limited our analysis to verifying whether the journals listed in ROAD were indexed in OpenAlex. We did not use OpenAlex's metadata for journal classification or other characteristics. As a result, any potential inaccuracies in OpenAlex's metadata, such as issues with document types or open access status, did not affect our results. This approach ensured that the quality of OpenAlex's metadata did not influence our analysis, as the focus was solely on checking the presence or absence of ROAD-listed OA journals in OpenAlex. For income data by country, we used the R package 'tmap' [22], which provides this information. Additionally, we enriched the metadata for 13 territories that were not listed in the 'World' dataset of tmap. Namely the

territories with the following ISO3 codes: BHR (Bahrain), BRB (Barbados), MLT (Malta), MUS (Mauritius), SGP (Singapore), SYC (Seychelles), GLP (Guadeloupe), GUF (French Guiana), REU (Réunion), MTQ (Martinique), GUM (Guam), MAC (Macau), HKG (Hong Kong).

## Methods

The study has a dual objective: (1) to analyse the overall coverage of OA journals in the three databases and (2) to investigate the geographical and disciplinary distribution to assess the extent to which disciplines, countries, and regions are represented in each database. For the analysis of overall coverage, we employed an UpSet graph to visualize the coverage and intersections among the three databases. This graphical representation allows for a comprehensive examination of the shared and unique journals indexed by each database. Regarding the distribution analysis, an indicator (the Coverage Index) was calculated to assess the representation of various entities (such as countries, continents, or disciplines) in each of the three databases (WoS, Scopus, and OpenAlex), relative to their representation in the entire ROAD database.

$$\text{Coverage Index}_{ij} = \frac{\left( \frac{\text{OA journals indexed}_{ij}}{\text{All OA journals indexed}_j} \right)}{\left( \frac{\text{OA journals indexed}_{i\ \text{ROAD}}}{\text{All OA journals indexed}_{\text{ROAD}}} \right)} \tag{1}$$

where $i$ the entity (continent, discipline, etc.) and $j$ the database (WoS, Scopus and OpenAlex). Specifically, this indicator compares the proportion of a given entity (e.g., a country) within the journals indexed by a particular database (e.g., OpenAlex) to the proportion of that entity within the entire ROAD database. For example, to calculate the indicator for Italy in OpenAlex, the proportion of Italy within the journals indexed by OpenAlex (1.6%) is divided by the proportion of Italy within the entire ROAD database (7%). The neutral value of this indicator is 1. Therefore, if a given country has an indicator of 1.30 in WoS, it would indicate that it is overrepresented by 30% in this database compared to the global structure of OA journals distribution within ROAD. This method allows for a refined assessment of the representation biases within each database, highlighting any discrepancies in the geographical and disciplinary distribution of OA journals across the three platforms.

## Results and discussion

As shown in Fig 2, among the journals indexed in the ROAD database, the WoS indexes 6,157 journals, while Scopus indexes 7,351, and OpenAlex indexes 34,217. A striking observation is the presence of 24,976 OA journals exclusively in OpenAlex, whereas only 182 journals are exclusively present in WoS and 373 in Scopus. Additionally, 4,094 OA journals are simultaneously indexed in all three databases, while 145 journals are indexed simultaneously in WoS and Scopus but not in OpenAlex.

The examination of continent representation in Scopus, the WoS, and OpenAlex reveals distinct trends, highlighting disparities in journal coverage on a global scale. As represented in Fig 3 below, all databases exhibit significant differences in continent representation. While Scopus and WoS generally show similar patterns, OpenAlex stands out with different representation trends. Scopus and WoS, despite minor variations in their continental indices, demonstrate a tendency towards over-representation of Oceania, North America and Europe, along with a relative under-representation of Africa and Asia. These databases seem to favor journal coverage from hegemonic regions, raising questions about equity and inclusivity in their indexing practices.

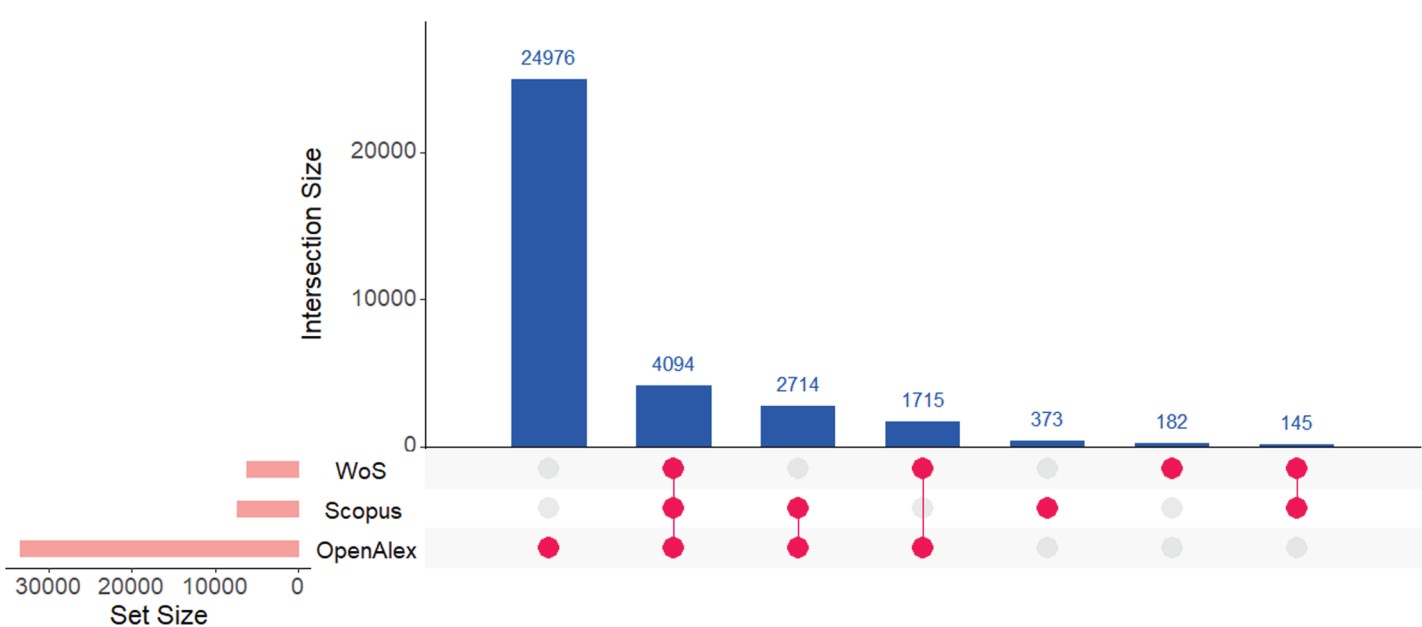

**Fig 2. Comparison of coverage of OA journals in Web of Science, Scopus, and OpenAlex.**

Conversely, OpenAlex presents more diverse representation patterns, with a trend towards better geographical equity. Although some disparities persist, such as the underrepresentation of Africa and Asia, OpenAlex appears to offer more inclusive coverage of journals from various regions worldwide.

The high values of Oceania in Scopus (3.39) and WoS (4.30) indicate that the region's presence in these databases is more than three and four times higher than the global average, respectively. This high representation is primarily driven by the significant contribution of Australia and New Zealand, both of which have robust research output and strong academic infrastructures. Their focus on publishing in high-impact, English-language journals aligns well with the indexing criteria of Scopus and WoS, leading to their disproportionate visibility. This imbalance may reflect a linguistic and systemic bias, where English-speaking countries with well-established research ecosystems are more prominently featured. Australia, in particular, largely benefited from the Regional Expansion of the WoS in 2006–2008 [23]. However, this is less the case in OpenAlex (1.46), which offers a more balanced representation of global research by including a broader range of OA sources, including those in multiple languages and from less prominent regions. Interestingly, Scopus and OpenAlex exhibit a common over-representation pattern regarding the coverage of South American OA journals, whereas OA journals from this continent are under-represented in the WoS.

The comparison of economic income groups (Fig 4) across Scopus, WoS, and OpenAlex databases reveals also notable disparities in representation. High-income economies, whether part of The Organisation for Economic Co-operation and Development (OECD) or not, are consistently over-represented in Scopus and WoS, reflecting historical biases. Unlike the WoS and Scopus, OpenAlex operates under a fundamentally different paradigm, prioritizing an open and inclusive indexing strategy. While the WoS and Scopus often rely on selective curation processes that may favour established publishers and higher-income regions, OpenAlex harvesting strategy embraces a broader, more decentralized approach. This allows for a wider representation of research outputs across various economic contexts, not necessarily through

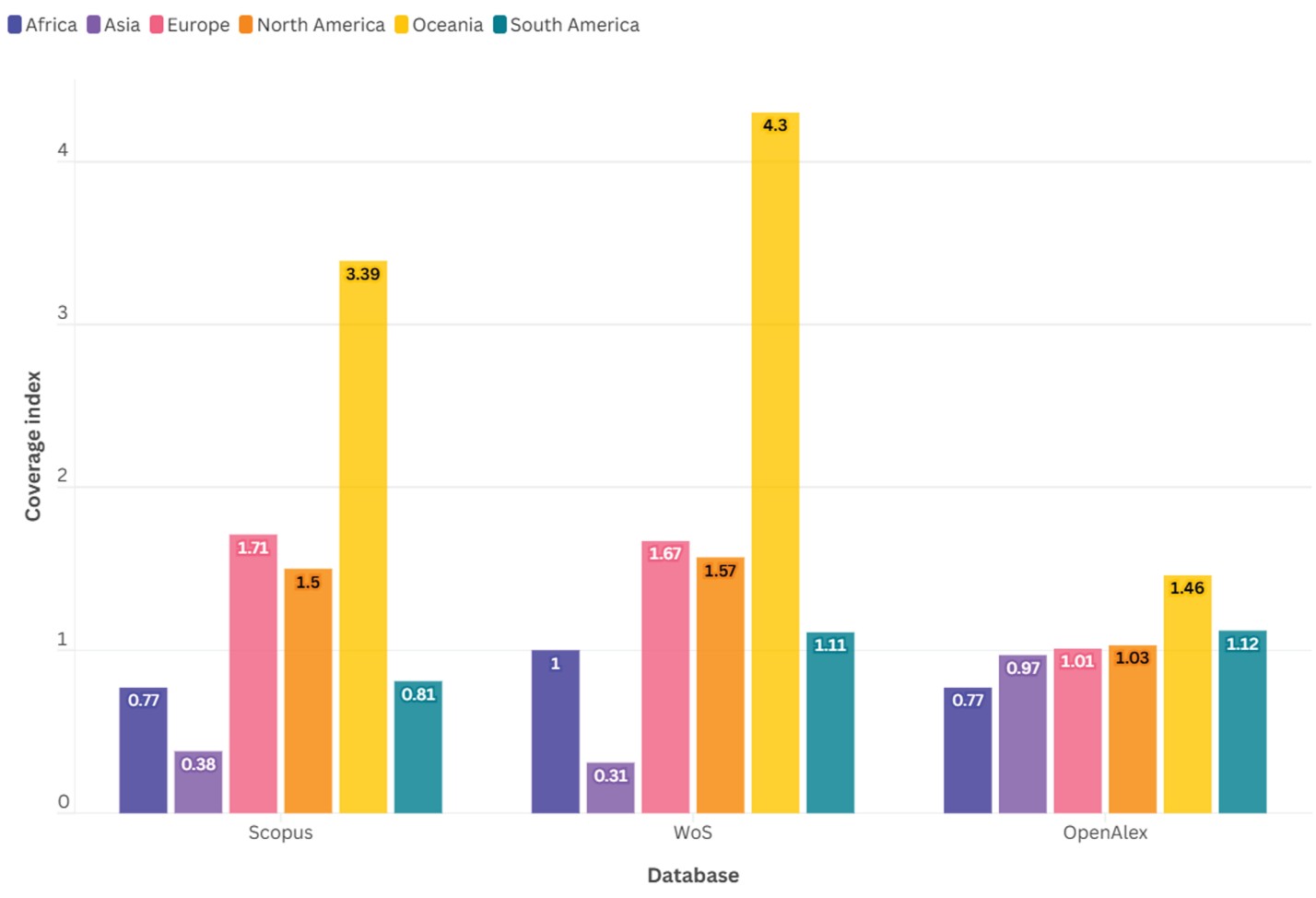

**Fig 3. Coverage index of databases by continent (neutral value = 1).**

targeted efforts but rather through a strategic commitment to openness and accessibility that contrasts with the more traditional, restrictive models of other platforms. Middle-income economies show varying levels of representation, with indicators fluctuating across databases.

Lower middle-income economies are particularly underrepresented in Scopus and WoS, highlighting systemic biases within these databases. OpenAlex presents a more equitable representation across middle-income categories. Low-income economies face consistent underrepresentation in Scopus and WoS, indicative of broader challenges in access to scholarly resources. Analysing the representation of disciplines across Scopus, WoS, and OpenAlex databases provides insights into the distribution of scholarly knowledge and potential biases within each database. Fig 5 comparing disciplinary representation across these databases reveals notable variations in coverage, reflecting broader trends in academic publishing and bibliographic indexing practices.

Scopus and WoS demonstrate unsurprising consistent biases towards certain disciplines, particularly within STEM fields such as Physics and Natural Sciences. International

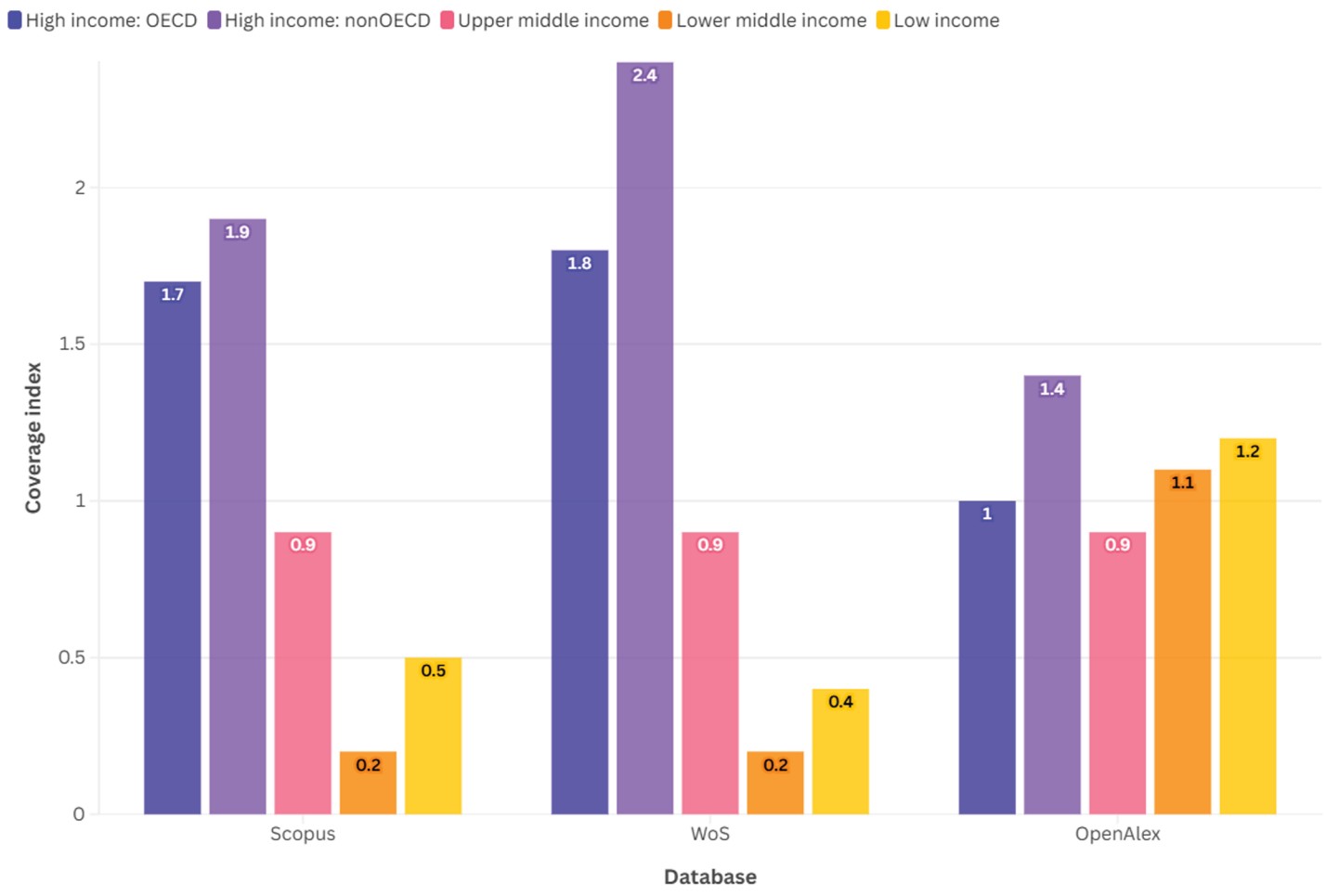

**Fig 4. Coverage index by database for different income groups (neutral value = 1).**

databases often use journal impact factors as a key criterion for inclusion, which may influence the prominence of quantitative research and publications from highly ranked institutions [24]. Consequently, disciplines within the humanities and social sciences may be under-represented in Scopus and WoS, reflecting historical publishing and citation biases. Conversely, OpenAlex presents a more inclusive approach to indexing, aiming to encompass a diverse range of scholarly outputs across disciplines. While OpenAlex exhibits slightly lower representation in some disciplines compared to Scopus and WoS, such as Applied Sciences, Medicine, and Technology, it offers a more balanced representation overall. This suggests that OpenAlex's indexing practices may be more reflective of the diverse disciplinary landscape and less influenced by traditional biases prevalent in academic publishing. Notably, academic subjects such as 'Social sciences' demonstrate higher representation in OpenAlex compared to Scopus and WoS, highlighting potential differences in indexing criteria and inclusivity across databases. This variation underscores the crucial importance of considering multiple databases to ensure comprehensive coverage across diverse disciplinary areas.

Figs 6–11 highlight the differences in the coverage of OA journals between the WoS, Scopus, and OpenAlex databases at the country level. The world maps (Figs 6, Fig 8, and Fig 10) represent the geographical distribution of the Coverage index for each database (described in

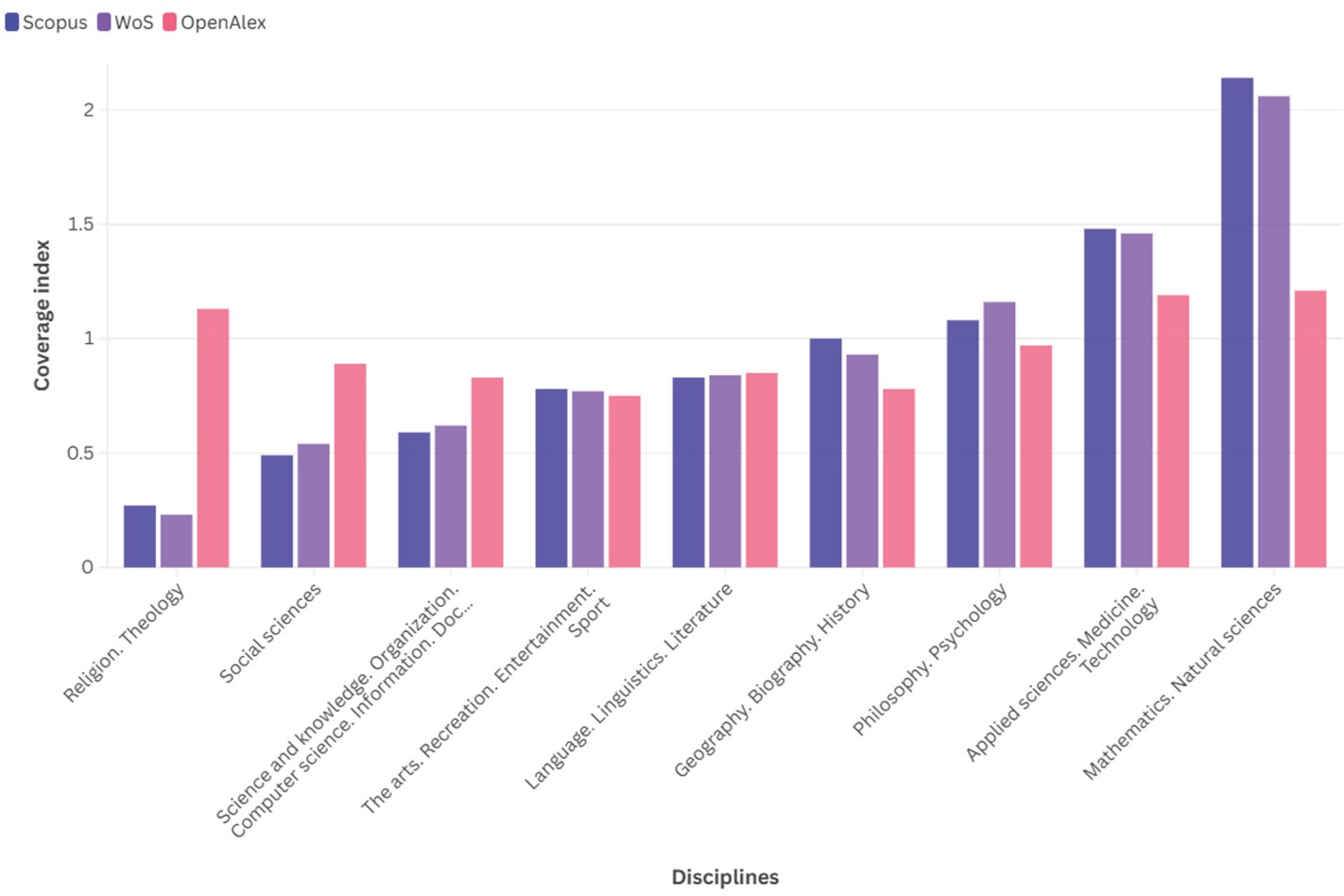

**Fig 5. Coverage index of databases by discipline (neutral value = 1).**

the Methods section), showcasing well-represented regions and countries as well as underrepresented ones. The circle packings (Fig 7, Fig 9, and Fig 11) show the share of ROAD journals covered by each database (colour gradient) whereas the size of the circles depend on the absolute number of ROAD journals per country. These representations allow us to highlight how significant the similarities between WoS, Scopus, and OpenAlex are.

Upon examining Figs 6–11, it becomes evident that certain regions, such as Western Europe and North America, are well represented in all three databases, with high Coverage index values. However, significant disparities emerge for other parts of the world. For instance, some areas in Africa, Asia, and Latin America exhibit relatively low coverage indices in the WoS and Scopus but better representation in OpenAlex. This difference can be attributed to OpenAlex's more inclusive indexing policy, which covers a broader range of OA journals from diverse regions around the world. In contrast, the WoS and Scopus may exhibit geographic biases, often favouring journals published in specific countries or regions, and applying selection processes that emphasize certain quality standards, which can further limit the diversity of represented sources. OpenAlex provides a more inclusive and balanced indexing of OA journals compared to both WoS and Scopus, which show similar coverage indices and comparable geographic biases. At the same time, it is worth noting that each database

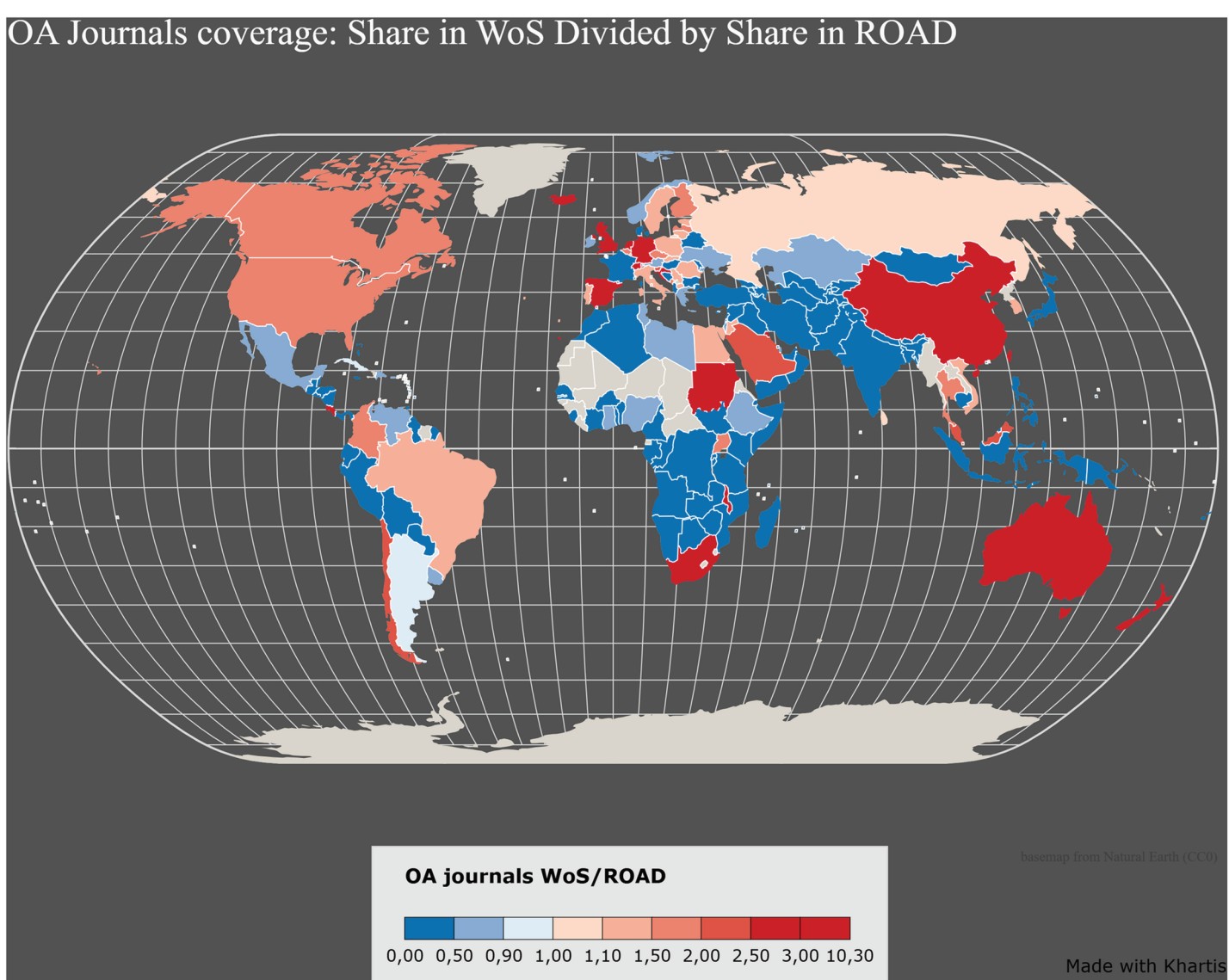

**Fig 6. Coverage index by country in the WoS.**

has its own strengths and weaknesses, and the choice depends on the specific needs of each research endeavour.

Our analysis also reveals two outliers we can interestingly focus on. In Europe, the case of France stands out as a notable exception. Despite its status as a high income country with a substantial scholarly output, French OA journals are consistently underrepresented across all three databases. This discrepancy raises questions about the systemic factors contributing to the exclusion of French journals from mainstream bibliographic databases. However, there is a possibility of French journals being disproportionately represented in the ROAD database, the reference database used for comparison. This potential over-representation in ROAD could skew perceptions of under-representation in Scopus, WoS, and OpenAlex. Further investigation into this discrepancy is necessary to understand the underlying factors contributing to the exclusion or under-representation of French OA journals.

Share of ROAD journals included in the Web of Science (%).
The circles' size is proportionnal to the absolute number of ROAD journals per country.

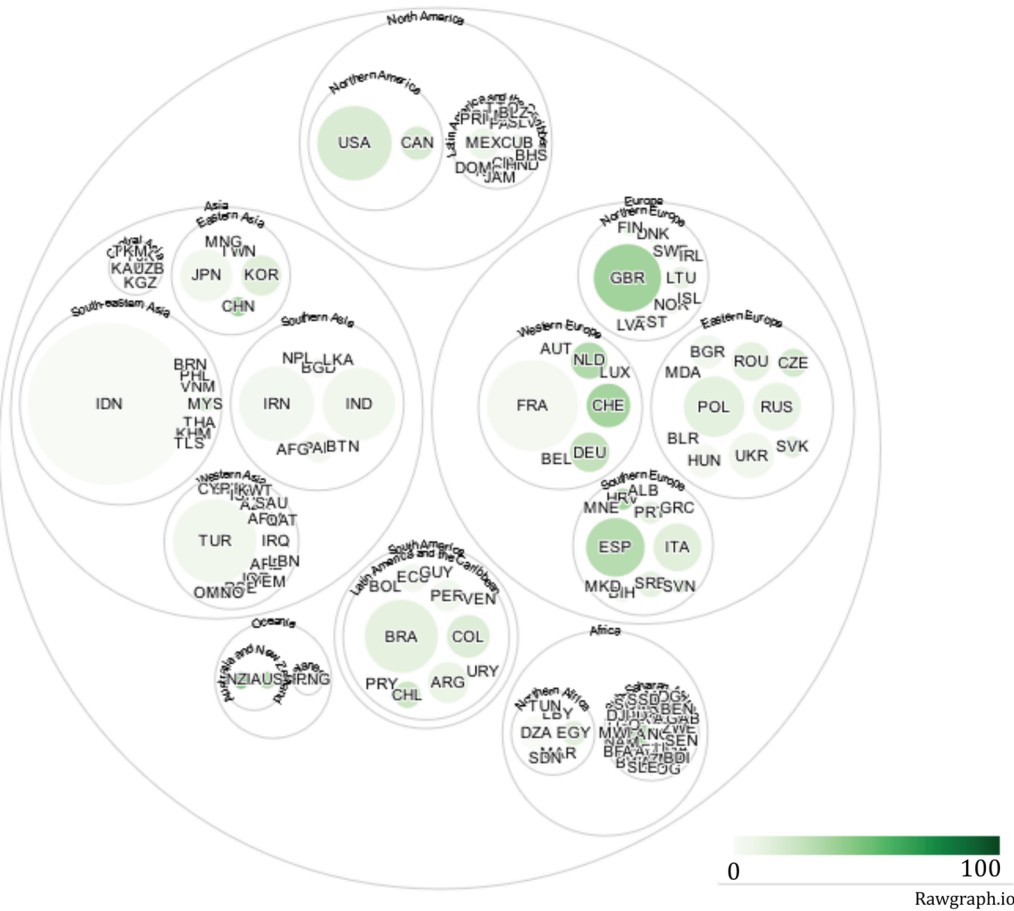

**Fig 7. Share of ROAD journals included in the WoS.**

The case of Indonesia also raises similar questions. Specifically, both OpenAlex and ROAD exhibit a notable pattern for this country. Indonesia emerges as the top contributor to the ROAD database (it represents the biggest circle on the circle packing plots). One possible explanation is that, since 2019, the Indonesian government has mandated through legislation that universities must create OA journals [25]. This policy has significantly increased the number of Indonesian journals in ROAD, potentially leading to an apparent over-representation in this database. As for France, this scenario necessitates further investigation to understand the impact of such policies on the visibility and representation of Indonesian journals in bibliographic databases like OpenAlex.

To sum up, this study presents a comprehensive assessment of the coverage of OA journals across three major scholarly databases: the WoS, Scopus, and OpenAlex. The analysis reveals significant variations in representation across different countries, regions, continents, and income groups, highlighting the importance of database selection when it comes to scholarly research and outputs. These results underscore the important role of OpenAlex

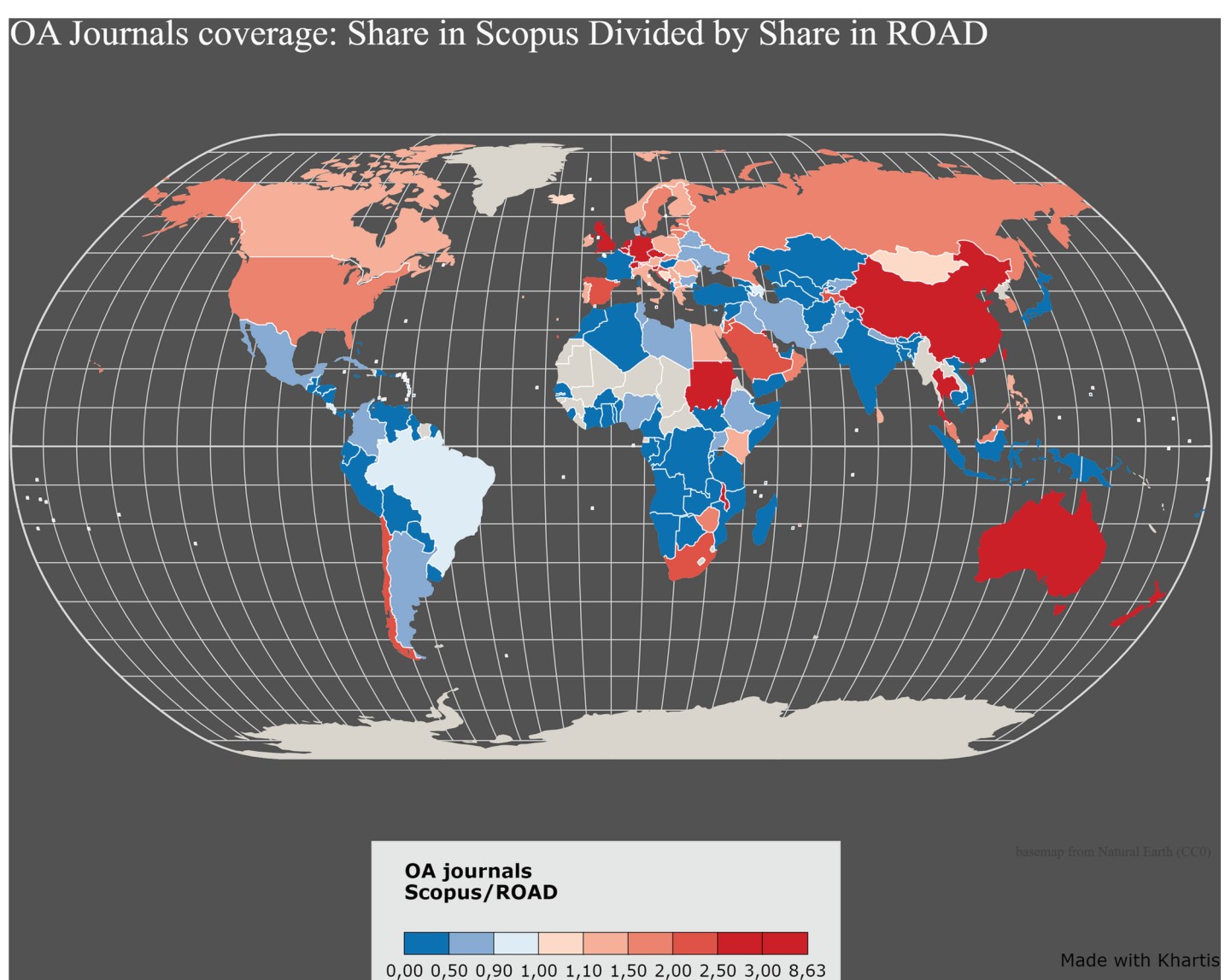

**Fig 8. Coverage index by country in the Scopus.**

in expanding the coverage of OA journals, with a significant proportion of journals exclusive to this database. However, it is crucial to acknowledge that some journals may publish a small number of publications, as evidenced by a recent study focusing on Diamond journals [26,27].

The examination of coverage indices highlights notable differences in the representation of countries and regions and disciplines across the three databases. While certain areas, such as Western Europe and North America, enjoy robust coverage across all platforms, disparities are evident in other regions. OpenAlex confirms its agenda and exhibits a more inclusive approach, capturing a broader range of OA journals from diverse geographic locations compared to WoS and Scopus. This suggests that OpenAlex serves as a valuable resource for researchers seeking comprehensive coverage, especially for studies focusing on underrepresented regions (as soon as the outliers are known, i.e., the case of Indonesia).

Share of ROAD journals included in Scopus (%).
The circles' size is proportionnal to the absolute number of ROAD journals per country.

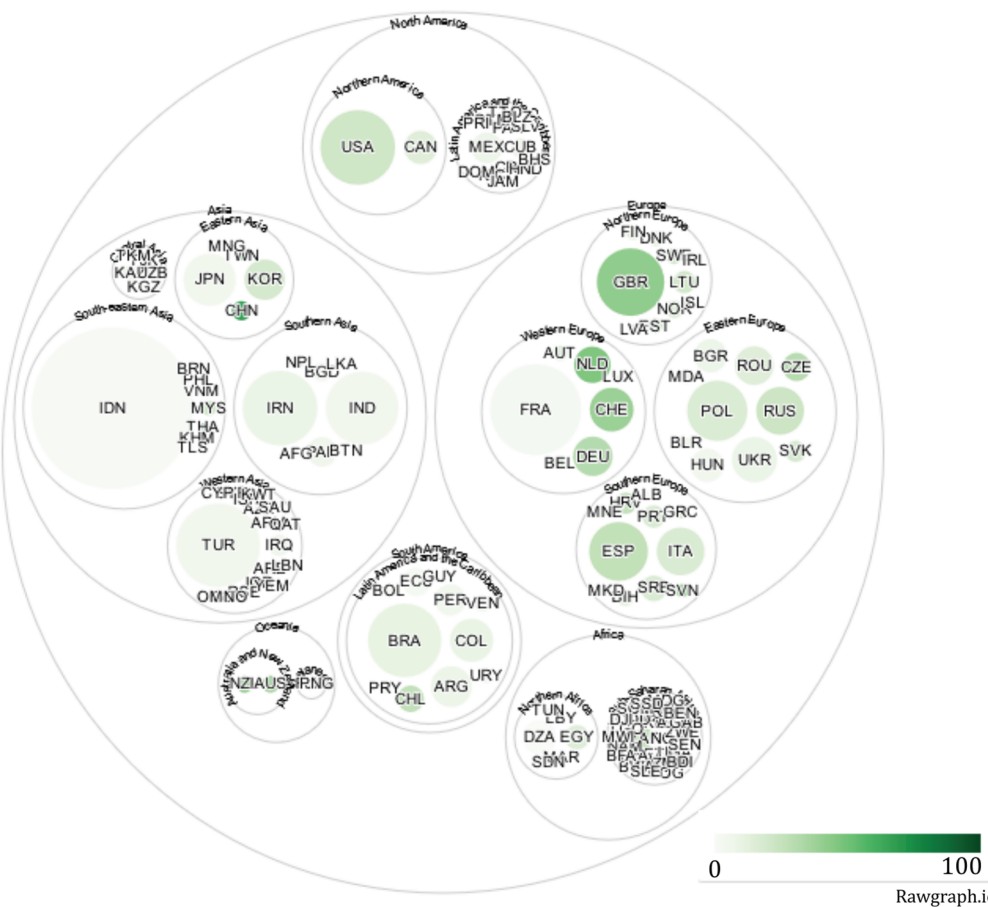

**Fig 9. Share of ROAD journals included in Scopus.**

Furthermore, the analysis points out disparities in coverage based on certain regions. In particular, Africa and parts of Asia tend to have lower coverage indices in WoS and Scopus compared to OpenAlex. This latter demonstrates a more equitable representation of journals from these regions, offering researchers access to a wider array of scholarly content. Likewise, analysis based on income groups further highlights disparities in database coverage. Low and middle-income countries often experience lower representation in WoS and Scopus compared to high-income countries. OpenAlex demonstrates a more equitable distribution of journals across income groups. Moreover, our findings underscore the importance of database selection in shaping scholarly research and knowledge dissemination. OpenAlex emerges as a new and valuable source for researchers seeking a more comprehensive coverage of OA journals, particularly from underrepresented regions, continents, development levels, and income groups.

In addition to the observed disparities in geographic, and income-based representation, the differences in database coverage can largely be attributed to the distinct criteria used by traditional databases such as WoS and Scopus for indexing journals. These databases often

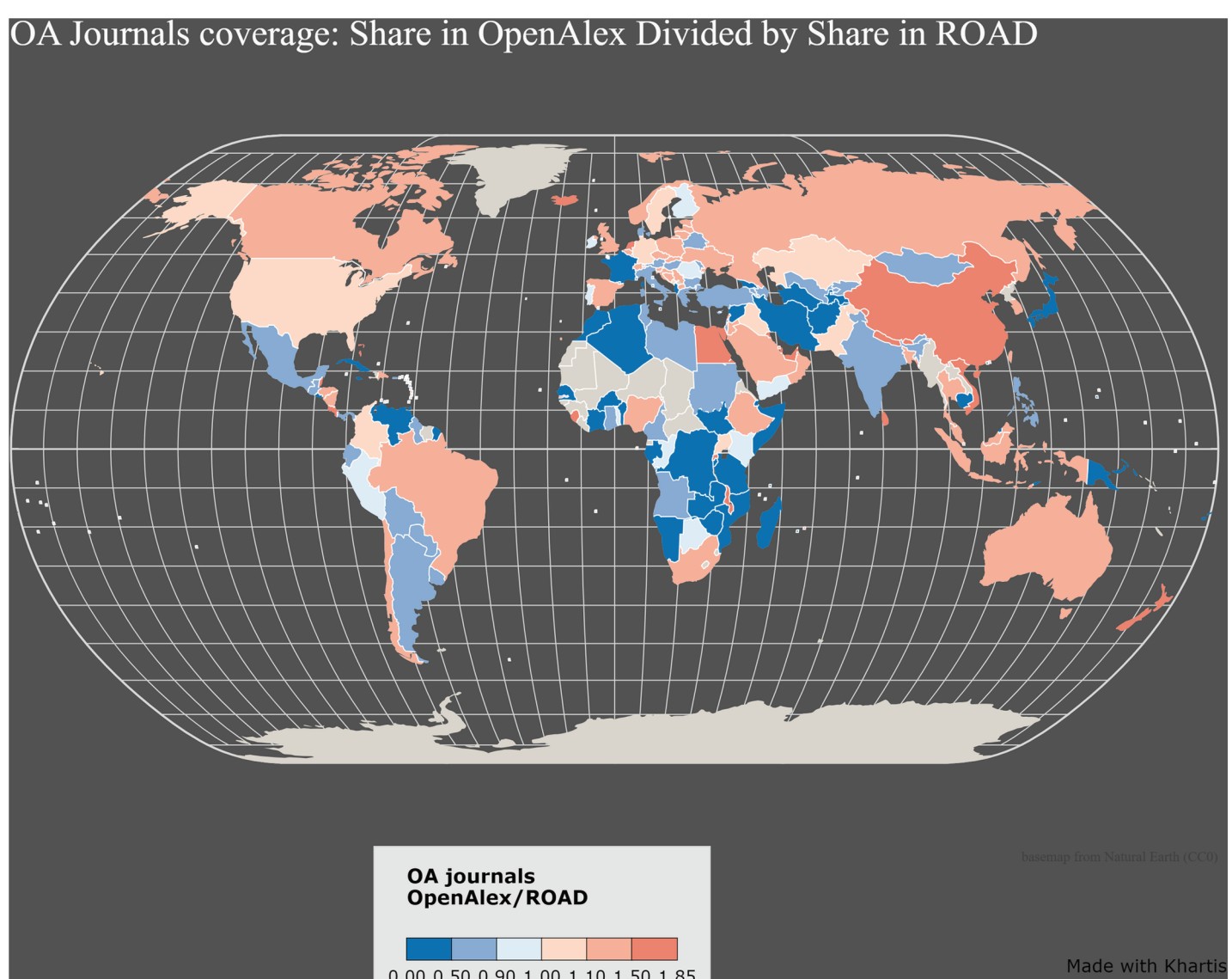

**Fig 10. Coverage index by country in OpenAlex.**

employ strict editorial and publishing standards, which, although ensuring a certain level of quality, may inadvertently exclude many open access journals, especially those from emerging or less-established regions. As a result, the WoS and Scopus are less inclusive in representing the full scope of scholarly output, particularly from non-Western countries or lower-income regions where open access models are rapidly growing. This exclusion is compounded by the fact that these traditional databases were initially designed around subscription-based journals, which have long dominated academic publishing. On the other hand, OpenAlex's inclusive approach to indexing OA journals, with fewer restrictions, enables it to better capture the diverse and evolving landscape of global scholarly communication, offering a more representative picture of the global research ecosystem. Thus, the selection and indexing practices of databases like WoS and Scopus can significantly shape the accessibility and visibility

Share of ROAD journals included in Open Alex (%).
The circles' size is proportionnal to the absolute number of ROAD journals per country.

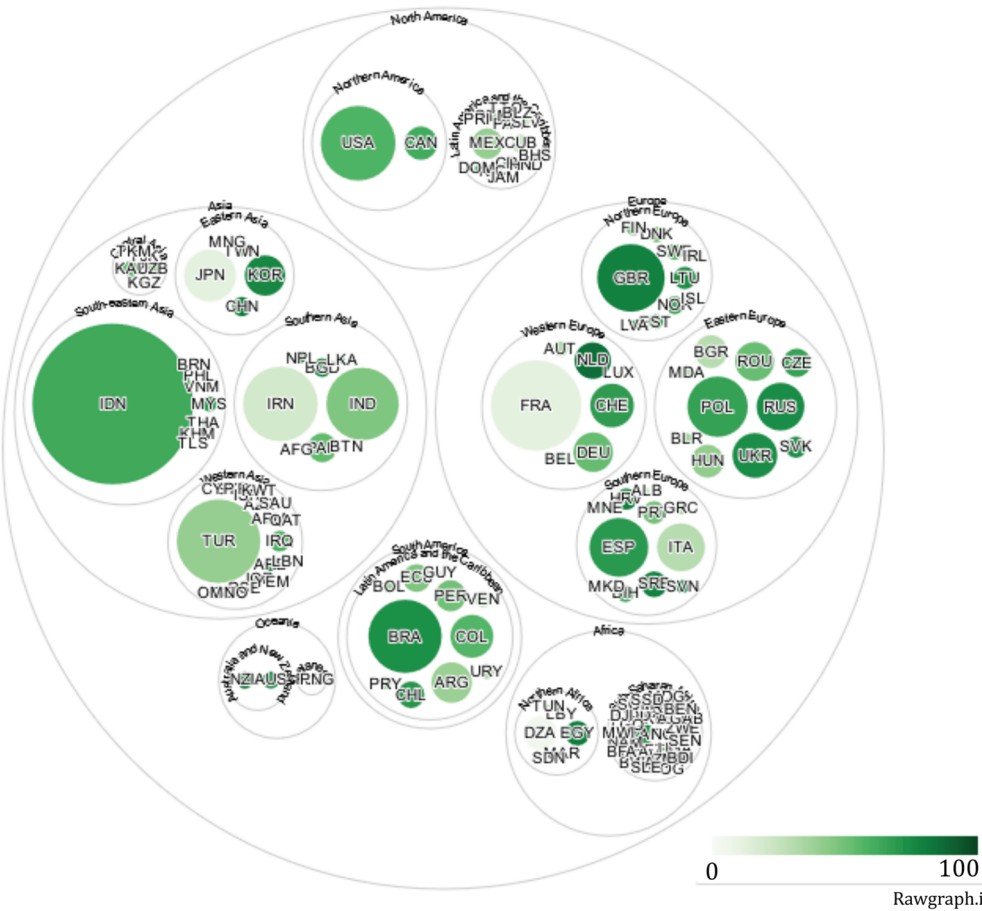

**Fig 11. Share of ROAD journals included in OpenAlex.**

of research from underrepresented regions, highlighting the critical role of database choice in academic research.

In discussing the notable cases of France and Indonesia in terms of OA journals and their respective under-representation in databases, it is essential to understand the factors that contribute to the high number of OA journals in these countries. Both France and Indonesia show notable differences in their representation of OA journals, and these discrepancies are influenced by various national and cultural factors. In France, the prominence of OA journals can be traced to several factors that have shaped the country's scholarly publishing landscape. One significant element is the long-standing tradition of 'revues de laboratoire' (journals issued from a specific research unit or a locally-based group), which are often founded, launched and managed by specific research laboratories, academic institutions, or specialized scientific communities mainly in Humanities and social sciences (HSS). These journals, mainly managed on a crafted and small-scale basis, although they may not always meet the standards required for indexing in Scopus or the WoS, represent a vital part of the French scholarly communication ecosystem. They contribute significantly to the high number of OA

journals in France, as many are supported by strong institutional backing and public funding. These journals provide an avenue for disseminating research and fostering open science within local or specialized communities. In addition, OpenEdition [28,29], a major French platform dedicated to scholarly publishing in the HSS, plays an important role in the French OA journal landscape and in the promotion of the Diamond OA. OpenEdition hosts a substantial number of French-language OA journals, especially in fields that are underrepresented in major international databases. It serves as a key infrastructure for the dissemination of scholarly knowledge in these fields. OpenEdition has helped increase the visibility and accessibility of French academic work, ensuring that journals, even if they are not indexed in large databases, contribute to the growing body of OA literature. This platform, combined with the French tradition of local journals, underscores the country's strong commitment to making academic research freely accessible.

In contrast, Indonesia has seen significant growth in OA journal production, driven largely by national policies and institutional efforts to increase the accessibility of research outputs. Government-led initiatives have encouraged universities and research institutions to embrace OA as part of their commitment to global academic visibility and knowledge sharing. Local OA journals have proliferated, supported by efforts to enhance the quality of Indonesian research and its international impact. Despite these advances, some challenges remain, such as limited resources for high-quality editorial management and insufficient international recognition, which may hinder these journals from meeting the criteria for indexing in global databases.

Both countries' OA journal landscapes reflect a combination of local initiatives and national policies. France benefits from a well-established tradition of 'revues de laboratoire' and platforms like OpenEdition, which contribute significantly to the country's OA journal vitality. Indonesia, while benefiting from more recent surges in OA journal production, has made significant progress through government support and institutional initiatives, though it faces challenges in establishing international recognition. These country-specific dynamics emphasize the importance of considering both national and regional factors when analyzing OA journals representation, as they play a significant role in shaping the openness, accessibility, and inclusivity of academic publishing.

## Conclusion

Our study highlights the role of database selection in shaping the landscape of scholarly research and knowledge dissemination. The disparities in coverage across the WoS, Scopus, and OpenAlex underscore the varying degrees of inclusion and representation of OA journals from different countries, continents, and income groups. OpenAlex, with its broader and more inclusive approach, emerges as a valuable resource for researchers, particularly those focusing on underrepresented regions or countries with lower income levels. This platform mitigates some of the biases inherent in traditional databases like WoS and Scopus, offering a more equitable distribution of OA journals.

To better understand our results on France under-representation in the WoS, Scopus and OpenAlex, a closer and more longitudinal examination is needed. Together with Indonesia, France is the country with the highest number of OA journals indexed in ROAD . Interestingly, the role of OA national policies has been important in the two countries, but OpenAlex unequally covers these countries (coverage index of 1.19 for Indonesia and 0.22 for France). Further investigations are needed to explore these differences.

## Limitations

While this study provides a comprehensive analysis of OA journal coverage across OpenAlex, WoS, and Scopus, it is essential to discuss certain limitations that may influence the interpretation of the results. First, metadata issues are a known challenge in OpenAlex, as well as in traditional databases like WoS and Scopus to a lesser extent. However, in this study, we minimized the impact of these potential biases by relying primarily on the metadata provided by the ROAD database, which is curated independently of OpenAlex. This approach allowed us to avoid many of the inaccuracies that could arise from OpenAlex's metadata, such as issues with the disciplinary classification or geographical indexation. Second, while ROAD is a robust resource for identifying OA venues, it is not without limitations too. Notably, over 2,000 journals in ROAD lacked disciplinary classifications, which may have affected the results presented in Fig 5 (disciplinary coverage by database). This could result in an incomplete representation of certain disciplines. Enhancing the metadata coverage for these journals would be an important step for future research aiming to refine the analysis. Finally, the absence of publication volume data for ROAD and the restricted availability of such data for WoS and Scopus limited our ability to analyze the impact of journal size on database representation. While publication thresholds could be explored using OpenAlex data, a more comprehensive investigation across all databases was beyond the scope of this study. Future work could address the identified gaps and further enhance the robustness of comparative analyses.

## Acknowledgments

The authors would like to thank ISSN International Centre for providing the ROAD data. We would like to express our heartfelt gratitude to Dr. Gaëlle BEQUET, Director of the ISSN International Centre, for reviewing the initial draft of this paper. Her insightful comments and constructive feedback have been invaluable in enhancing the quality and depth of our work. We sincerely appreciate her time and effort, which have significantly contributed to the improvement of this article.

## Author contributions

**Conceptualization:** Abdelghani Maddi, Marion Maisonobe, Chérifa Boukacem-Zeghmouri.

**Data curation:** Abdelghani Maddi, Marion Maisonobe.

**Formal analysis:** Abdelghani Maddi, Marion Maisonobe, Chérifa Boukacem-Zeghmouri.

**Investigation:** Abdelghani Maddi, Marion Maisonobe, Chérifa Boukacem-Zeghmouri.

**Methodology:** Abdelghani Maddi, Marion Maisonobe, Chérifa Boukacem-Zeghmouri.

**Project administration:** Abdelghani Maddi, Marion Maisonobe, Chérifa Boukacem-Zeghmouri.

**Software:** Abdelghani Maddi, Marion Maisonobe.

**Supervision:** Abdelghani Maddi, Marion Maisonobe, Chérifa Boukacem-Zeghmouri.

**Validation:** Abdelghani Maddi, Marion Maisonobe, Chérifa Boukacem-Zeghmouri.

**Visualization:** Abdelghani Maddi, Marion Maisonobe.

**Writing – original draft:** Abdelghani Maddi, Marion Maisonobe, Chérifa Boukacem-Zeghmouri.

**Writing – review & editing:** Abdelghani Maddi, Marion Maisonobe, Chérifa Boukacem-Zeghmouri.

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
