## [Decision Letter · Decision Letter 0]

29 Nov 2024

PONE-D-24-47603Geographical and Disciplinary Coverage of Open Access Journals: OpenAlex, Scopus and WoSPLOS ONE

Dear Dr. MADDI,

Thank you for submitting your manuscript to PLOS ONE. After careful consideration, we feel that it has merit but does not fully meet PLOS ONE’s publication criteria as it currently stands. Therefore, we invite you to submit a revised version of the manuscript that addresses the points raised during the review process. Please, consider carefully the rich and constructive comments of both reviewers for the revision of your paper.

We look forward to receiving your revised manuscript.

Kind regards,

Alberto Baccini, Ph.D.

Academic Editor

PLOS ONE

4. We note that Figure 5 in your submission contain [map/satellite] images which may be copyrighted. All PLOS content is published under the Creative Commons Attribution License (CC BY 4.0), which means that the manuscript, images, and Supporting Information files will be freely available online, and any third party is permitted to access, download, copy, distribute, and use these materials in any way, even commercially, with proper attribution. For these reasons, we cannot publish previously copyrighted maps or satellite images created using proprietary data, such as Google software (Google Maps, Street View, and Earth). For more information, see our copyright guidelines: http://journals.plos.org/plosone/s/licenses-and-copyright.

1. You may seek permission from the original copyright holder of Figure 5 to publish the content specifically under the CC BY 4.0 license. 

Reviewers' comments:

Reviewer's Responses to Questions

**Comments to the Author**

1. Is the manuscript technically sound, and do the data support the conclusions?

Reviewer #1: Yes

Reviewer #2: Yes

2. Has the statistical analysis been performed appropriately and rigorously? 

Reviewer #1: Yes

Reviewer #2: Yes

3. Have the authors made all data underlying the findings in their manuscript fully available?

Reviewer #1: No

Reviewer #2: Yes

4. Is the manuscript presented in an intelligible fashion and written in standard English?

Reviewer #1: Yes

Reviewer #2: Yes

5. Review Comments to the Author

Reviewer #1: Thank you for the opportunity to review the manuscript entitled “Geographical and Disciplinary Coverage of Open Access Journals: OpenAlex, Scopus, and WoS.” I have several recommendations that may help improve the study.

1) The manuscript lacks clarity in explaining the criteria for comparison between the databases. For instance, the authors do not clarify in the methodology how they addressed classification differences between ROAD and the other databases. What is the total number of open access journals included in ROAD? How were these journals selected? What filters, or additional methods, were applied in this selection process?

2) The Results and Discussion sections mention notable exceptions, such as France and Indonesia; however, there is no in-depth analysis of the factors underlying these differences, such as publication policies or national initiatives that could contribute to these disproportions.

3) In the literature review, the authors cite several studies that discuss limitations of OpenAlex, including metadata accuracy issues, such as the precision of document types and open access status. However, there is no detailed discussion of how these limitations may have affected the results. It would be beneficial to provide more information regarding potential biases that may arise from metadata inaccuracies in OpenAlex.

Reviewer #2: This study is a useful contribution to efforts evaluating the potential effectiveness of OpenAlex as a more comprehensive source of data, and as an improvement on traditional unrepresentative data sources like Web of Science and Scopus. Studies of this kind are essential as OpenAlex experiences massive uptake globally to provide users with a critical account of its coverage and idiosyncrasies.

I praise the authors for how well-written this piece is. They provide a useful overview of the complex and evolving publishing landscape, describe their methodology clearly, and convey key findings with emphasis and helpful brevity. It was a pleasure to read. In this light, my comments are largely observational which I invite them to consider or disregard at their and the editor's discretion.

1. The abstract notes the ROAD database indexes 62,701 OA active resources, which I eventually understand to be journals. I suggest using clearer terminology and staying consistent with journals, at least in the abstract, and otherwise explaining the term in the paper if it is necessary to retain it. I at first assumed resources to include a wider range of outputs or venues.

2. In the abstract, the word “striking” is used descriptively within one sentence of each other. I suggest removing one of them to avoid repetition, and particularly the second one as it reads a bit leading and more neutral language is favourable.

3. The second sentence in the introduction it feels amiss to not acknowledge economic or philosophical factors which spur the buy-in or championship of openness.

4. In the introduction, and in the discussion, I think WoS and Scopus’ indexing criteria could be referenced- their lag in terms of coverage and inclusivity is also largely due to their indexing standards of which OA sources are excluded for various reasons.

5. Third paragraph of the introduction—typo: “OpenAlex’s agenda”.

6. The following sentence might add an Oxford comma after “OA repositories” for flow and clarity.

7. The research questions are written to be future-oriented, implying a trajectory from findings. I recommend rephrasing as I do not think this paper can answer either question: whether OpenAlex overcomes disciplinary biases to ensure a more equitable representation of research is subject to more factors than coverage. I suggest reworking to something like: “Does OpenAlex possess broader disciplinary representation than traditional databases (WoS and Scopus)?

8. 5 publications seems like a very low threshold to consider a journal active: could the authors elaborate on why this was selected? I wonder how many of these appear in the WoS ESCI? This may provide some indication if they are emerging and acceptable based on a low publication count.

9. In the literature review you reference Alperin et al.’s (2024) noted a lack of studies on OpenAlex and its limitations but you proceed to review 10 studies on OpenAlex- though all are from 2023 and 2024, it feels perhaps less relevant to note this lack of literature so perhaps just keep the sentence focused on Alperin et al.’s other findings.

10. CNRS (French national research organism) – is this supposed to read organization?

11. Did the authors consider manually classifying the 2263 journals lacking a discipline based on the ROAD classification?

12. In the abstract you note that ROAD indexes 62,701 resources as of May 2024 but your data was extracted in October 2023- to benchmark the coverage with your study data I would suggest revising if you can find a more accurate ROAD indexing estimate to October 2023.

13. The UpSet graph formula seems to have a typo in its caption note where Scopus is repeated twice and should include OpenAlex.

14. After Figure 1 it is stated “However, it is crucial to acknowledge that some journals may publish a small number of publications, as evidenced by a recent study focusing on Diamond journals (Bosman et al., 2021).” Relating back to your publication threshold, it would be interesting to compare which exclusive or cross-indexed sources have a high threshold, say, above 200 publications, and above 400 publications. This would hint at established or enduring sources vs. smaller or emerging journals like community run diamond ones.

15. More might possibly be said about Asia’s underrepresentation in OpenAlex as well- particularly in relation to the scholarly publishing landscape and mandates in China and Indonesia's high contribution.

16. Figure 2 would benefit from axis labels.

17. The findings and discussion are conflated as results are engaged with more than descriptively—I recommend either separating the two or renaming the Results section to “Results and Discussion”.

18. Figure 3 could have the indication of (neutral value = 1 in its title).

19. The research objectives or questions could state that you also look at income groups, perhaps grouped into region, this economic geographical aspect could be positioned earlier on.

20. Include the full name of the OECD when it is introduced.

21. Figure 4 would also benefit from axis labels. Overall, the figures are quite nice but could be aligned in design, positioning of legends, and labelling to make them look more coordinated/uniform.

22. In your conclusion I suggest adding a Limitations section. Your literature review finds that metadata issues are abundant in OpenAlex, as well as a prevalent feature of WoS and Scopus- comment on how this may affect your findings. Similarly, the ROAD database which lacked disciplinary classifications for over 2000 journals may have affected your Figure 4 results.

23. Figure 5 is quite busy and possibly ineffective due to the overlapping nature of the labels. I’m not sure it adds much more to the maps. Perhaps consider just keeping the map coverage or providing descriptive information about the top countries for each database.

6. PLOS authors have the option to publish the peer review history of their article (what does this mean?). If published, this will include your full peer review and any attached files.

Reviewer #1: No

Reviewer #2: No

---

## [Author Response · Author response to Decision Letter 1]

31 Dec 2024

Dear Editor and Reviewers,

We sincerely appreciate the time and effort you have dedicated to reviewing our manuscript and providing such constructive feedback. We have carefully considered all your comments and suggestions and have revised the manuscript accordingly. Below, we provide a point-by-point response to the comments raised by both the editor and reviewers. We hope these revisions address your concerns and improve the overall quality and clarity of the manuscript.

Journal Requirements:

1. PLOS ONE Style Requirements

We have reformatted the paper according to PLOS ONE's style requirements. We hope that this meets your expectations.

2. Data Availability Statement

Thank you for pointing this out. The data are available under the CC BY license on Zenodo: https://zenodo.org/records/14389358.

3. PLOS LaTeX Template

We have reformatted the paper according to PLOS ONE's style requirements. We hope that this meets your expectations.

4. Figure 5 Copyright Concerns

Thank you for your comment and the detailed information provided. We designed all the maps in Figure 5 (in the new version, figures 6, 8 and 10) using the “Khartis” software, an open-source tool that uses Natural Earth base maps under a CC0 license. Thus, the base maps comply with the CC BY 4.0 requirements.

Reviewers' Comments:

Reviewer #1:

1. Criteria for Comparison Between Databases

Thank you for pointing this out. We have added the following clarification to the methodology:

“For the purpose of this study, ROAD was used as the gold standard for comparing the coverage of OpenAlex, Scopus, and WoS in Open Access (OA) journals. ROAD (Directory of Open Access Scholarly Resources) includes exclusively OA journals and provides comprehensive metadata for each journal, including ISSN, EISSN, country, and discipline classification. We included all journals from ROAD with complete metadata, without applying any additional filters. The data extraction from ROAD was conducted in October 2023, yielding 253,200 identifiers (ISSN and/or EISSN) for 183,158 distinct journals.

For our analysis, we focused on active OA journals in OpenAlex that had more than five publications. From the total of 183,158 journals in OpenAlex, 6,632 journals with five or fewer publications were excluded from the analysis. Consequently, we considered 176,526 active journals from OpenAlex. Additionally, we included 29,262 active journals from Scopus and 23,189 active journals from the WoS Core Collection (covering AHCI, SSCI, SCIE, and ESCI).

In addressing classification differences, we used the metadata provided by ROAD for the country and discipline classification, as ROAD’s classifications are considered reliable, being provided by the ISSN Centre. This approach allowed us to bypass the issue of reconciling different classification systems between the databases. Our comparison therefore focused on the indexing of OA journals listed in ROAD across the other databases (OpenAlex, WoS, and Scopus).”

2. Notable Exceptions: France and Indonesia

Thank you for pointing this out. We have added the following discussion:

“In discussing the notable cases of France and Indonesia in terms of OA journals and their respective under-representation in databases, it is essential to understand the factors that contribute to the high number of OA journals in these countries. Both France and Indonesia show notable differences in their representation of OA journals, and these discrepancies are influenced by various national and cultural factors.

In France, the prominence of OA journals can be traced to several factors that have shaped the country’s scholarly publishing landscape. One significant element is the long-standing tradition of “revues de laboratoire” (journals issued from a specific research unit or a locally-based group), which are often founded, launched and managed by specific research laboratories, academic institutions, or specialized scientific communities mainly in Humanities and social sciences. These journals, mainly managed on a crafted and small-scale basis, although they may not always meet the standards required for indexing in Scopus or the WoS, represent a vital part of the French scholarly communication ecosystem. They contribute significantly to the high number of OA journals in France, as many are supported by strong institutional backing and public funding. These journals provide an avenue for disseminating research and fostering open science within local or specialized communities. In addition, OpenEdition (26,27), a major French platform dedicated to scholarly publishing in the Humanities and social sciences, plays an important role in the French OA journal landscape and in the promotion of the Diamond OA. OpenEdition hosts a substantial number of French-language OA journals, especially in fields that are underrepresented in major international databases. It serves as a key infrastructure for the dissemination of scholarly knowledge in these fields. OpenEdition has helped increase the visibility and accessibility of French academic work, ensuring that journals, even if they are not indexed in large databases, contribute to the growing body of OA literature. This platform, combined with the French tradition of local journals, underscores the country's strong commitment to making academic research freely accessible.

In contrast, Indonesia has seen significant growth in OA journal production, driven largely by national policies and institutional efforts to increase the accessibility of research outputs. Government-led initiatives have encouraged universities and research institutions to embrace OA as part of their commitment to global academic visibility and knowledge sharing. Local OA journals have proliferated, supported by efforts to enhance the quality of Indonesian research and its international impact. Despite these advances, some challenges remain, such as limited resources for high-quality editorial management and insufficient international recognition, which may hinder these journals from meeting the criteria for indexing in global databases.

Both countries’ OA journal landscapes reflect a combination of local initiatives and national policies. France benefits from a well-established tradition of “revues de laboratoire” and platforms like OpenEdition, which contribute significantly to the country’s OA journal vitality. Indonesia, while benefiting from more recent surges in OA journal production, has made significant progress through government support and institutional initiatives, though it faces challenges in establishing international recognition. These country-specific dynamics emphasize the importance of considering both national and regional factors when analyzing OA journals representation, as they play a significant role in shaping the openness, accessibility, and inclusivity of academic publishing.”

3. Metadata Accuracy Issues in OpenAlex

Thank you for your comment. To address this concern, we clarified in the data section:

“It is important to note here that our methodology was specifically designed to avoid potential biases related to metadata inaccuracies in OpenAlex. By relying on ROAD as the gold standard, which provides comprehensive and reliable metadata for OA journals, we limited our analysis to verifying whether the journals listed in ROAD were indexed in OpenAlex. We did not use OpenAlex’s metadata for journal classification or other characteristics. As a result, any potential inaccuracies in OpenAlex's metadata, such as issues with document types or open access status, did not affect our results. This approach ensured that the quality of OpenAlex’s metadata did not influence our analysis, as the focus was solely on checking the presence or absence of ROAD-listed OA journals in OpenAlex.”

Reviewer #2:

Reviewer #2: This study is a useful contribution to efforts evaluating the potential effectiveness of OpenAlex as a more comprehensive source of data, and as an improvement on traditional unrepresentative data sources like Web of Science and Scopus. Studies of this kind are essential as OpenAlex experiences massive uptake globally to provide users with a critical account of its coverage and idiosyncrasies.

I praise the authors for how well-written this piece is. They provide a useful overview of the complex and evolving publishing landscape, describe their methodology clearly, and convey key findings with emphasis and helpful brevity. It was a pleasure to read. In this light, my comments are largely observational which I invite them to consider or disregard at their and the editor's discretion.

###

Thank you for reading our paper and the general appreciation and constructive feedback. We tried to address all the comments raised and we hope that the new version is satisfactory.

1. Terminology in Abstract

Thank you for pointing this out. The ROAD database primarily indexes OA journals, but it also includes other types of open access resources, such as monographs in series, which are also considered part of the open access publishing landscape. To clarify this distinction, we have added a figure (Figure 1) to emphasize that almost all the resources in ROAD are journals. Additionally, we have decided to use the term “journal” throughout the paper, as these resources follow journal standards in terms of editorial processes and article publication. This approach aims to provide greater consistency and clarity, particularly in the abstract, where the term "journal" will be used exclusively to refer to all types of resources indexed in ROAD.

2. Repetition of “Striking” in Abstract

Thank you for this remark. We have removed the second instance of “striking” to avoid redundancy.

3. Economic and Philosophical Factors in Open Access Transition

Thank you for your suggestion. We have expanded the introduction to include economic and philosophical drivers of open access:

“This transition is driven by policy mandates, institutional initiatives, and changing attitudes towards scholarly communication, alongside economic considerations, such as the rising costs of subscription-based models, and philosophical principles promoting equity and accessibility in scholarly publishing, with many arguing that publicly funded research should be freely accessible to all.”

4. Indexing Criteria of WoS and Scopus

Thank you for pointing this out. We have added a discussion of WoS and Scopus’ indexing criteria in both the introduction and discussion sections, emphasizing how these criteria may exclude many OA journals, particularly from non-Western countries.

5. Typo in Introduction: "OpenAlex’s agenda"

Thank you. It has been corrected.

6. Oxford Comma After “OA Repositories”

Thank you. We added an Oxford comma after OA repositories.

7. Rephrasing Research Questions

Thank you for pointing this out and for the suggestion. We have used the proposed rephrasing of the research question to "Does OpenAlex possess broader disciplinary representation than traditional databases (WoS and Scopus)?"

8. Threshold for Active Journals

Thank you for your comment regarding the threshold of 5 publications to consider a journal as "active." We selected this threshold after careful consideration and with the aim of applying more objective criteria for identifying active journals. One reason for this choice is to balance the inclusion of emerging journals while still filtering out very low-activity sources. We also considered other factors, such as annual publication frequency, to ensure that a journal was genuinely active within a given timeframe.

Additionally, we attempted to retrieve the most up-to-date data from OpenAlex, downloading it again on December 11, 2025, to check for journal activity. Unfortunately, the quality of the data from OpenAlex turned out to be quite poor (especially the number of publications by journal by year), which limits the reliability of some of the information we have gathered. This, of course, impacted our ability to set more robust and detailed criteria for active journals, especially compared to other databases like Web of Science (WoS) and Scopus, where the data is more structured and reliable. For WoS and Scopus, we worked with data that is already limited to journals indexed as "active."

The threshold of 5 publications, while not perfect, was inspired by a recent study presented at STI2024 by Diego Chavarro and Juan Pablo Alperín, titled "Equity in Scholarly Visibility: Bridging the Gap for Journals using Open Journal Systems in OpenAlex," where they used 1 publication as a threshold.

We have added this point regarding the volume of publications and the limitations of the OpenAlex data to the limitations section.

9. Sentence on Alperin et al. (2024)

Thank you for highlighting this. We removed the sentence "but noted a lack of studies focusing on OpenAlex itself and its limitations."

10. CNRS Acronym Correction

Thank you for this remark. We corrected the development of the CNRS acronym by the right translation: the "French national center for scientific research" as we can find on their website www.cnrs.fr.

11. Manual Classification of Journals

Thank you for this comment. Indeed, we wanted to classify manually the resources lacking a discipline, but we didn’t to avoid introducing an arbitrary classification different from that of ROAD. The second reason is the lack of time and resources. But if the reviewer thinks that it’s necessary, we are willing to enrich the disciplinary data manually. In the meantime, we added this point as a limitation of our study.

12. ROAD Data Benchmarking

Thank you for this suggestion. Actually, we didn’t extract the ROAD data (they are not available online); the data was kindly sent in an XML file by the database provider. So, we could ask the ISSN center if they could provide us with the data from October 2023, if the reviewer thinks it’s necessary.

13. UpSet Graph Caption

Thank you for this remark. However, at least in this new version, this typo is not present.

14. Publication Volume Analysis

Thank you for this valuable suggestion. Analyzing the publication volume of journals, particularly comparing those with high publication thresholds (e.g., above 200 or 400 publications) to smaller or emerging journals like community-run Diamond ones, would indeed provide interesting insights. Unfortunately, our current dataset does not include the number of publications for ROAD, WoS, and Scopus-listed journals. For ROAD, only metadata about journal presence is available, and for WoS and Scopus, we have access solely to their journal lists, without detailed publication counts. However, for OpenAlex, where publication data is available, we could perform such an analysis if desired, but we are afraid that the accuracy of the publication data per journal in OpenAlex is still not good enough to perform such an analysis. If you believe this analysis would add value to the study, we would be happy to include it in a revised version of the manuscript. We added this point in the limitations section.

15. Asia’s Underrepresentation

Thanks for reporting this. We have responded to this comment by addressing the first reviewer's 2nd comment.

16. Axis Labels in Figure 2

Thank you. It’s done.

17. Renaming Results Section

Thank you. It’s done. We’ve renamed the Results section to "Results and Discussion."

18. Neutral Value Indication in Figure 3

Thank you. It’s done.

19. Research Objectives and Income Groups

Thank you for pointing this out. We have added a new research question (How do traditional and alternative scholarly databases differ in their representation of OA journals across macro regions and income groups?) and reordered the question to respect the paper’s structure.

20. Full Name of OECD

Thank you. It’s done.

21. Figure 4 Improvements

Thank you. It’s done.

22. Limitations Section

Thank you for this suggestion regarding the inclusion of a Limitations section. We have now added the following section:

"While this study provides a comprehensive analysis of OA journal coverage across OpenAlex, WoS, and Scopus, it is essential to

---

## [Decision Letter · Decision Letter 1]

30 Jan 2025

PONE-D-24-47603R1Geographical and Disciplinary Coverage of Open Access Journals: OpenAlex, Scopus and WoSPLOS ONE

Dear Dr. MADDI,

Thank you for submitting your manuscript to PLOS ONE. After careful consideration, we feel that it has merit but does not fully meet PLOS ONE’s publication criteria as it currently stands. Therefore, we invite you to submit a revised version of the manuscript that addresses the points raised during the review process.

The editorial check individuated some minor issues regarding statements that seem to be unsupported/sufficiently referenced. They are as follows:

Line 281: "Australia, in particular, largely benefited from the Regional Expansion of the WoS in 2006-2008." - this probably requires a reference;

Line 278 - 280: We believe this sentence may be written too strongly and and not be fully supported by the methodology/results. We suggest rewriting to something to the effect of "This imbalance may reflect a linguistic and systemic bias, where English-speaking countries with well-established research ecosystems are more prominently featured". As academic editor, I suggest to accept the suggestion of the editorial staff, or, alternatively, to add references justyfying your stronger statement.

Line 313 - 315 "These databases prioritize journals with high impact factors, often favoring quantitative research outputs and publications from well-established and high-ranked academic 314 institutions." - this statement appears unsupported and would require either a supporting reference or rephrasing. 

line 254 says "A striking observation is the presence of 25,658 OA journals exclusively in OpenAlex". However the figure 2 shows 24976.

line 256 says "Additionally, 4,104 OA journals are simultaneously indexed in all three databases". However the figure 2 shows 4094.

We look forward to receiving your revised manuscript.

Kind regards,

Alberto Baccini, Ph.D.

Academic Editor

PLOS ONE

Journal Requirements:

Reviewers' comments:

Reviewer's Responses to Questions

**Comments to the Author**

1. If the authors have adequately addressed your comments raised in a previous round of review and you feel that this manuscript is now acceptable for publication, you may indicate that here to bypass the “Comments to the Author” section, enter your conflict of interest statement in the “Confidential to Editor” section, and submit your "Accept" recommendation.

Reviewer #1: All comments have been addressed

Reviewer #2: All comments have been addressed

2. Is the manuscript technically sound, and do the data support the conclusions?

Reviewer #1: Yes

Reviewer #2: Yes

3. Has the statistical analysis been performed appropriately and rigorously? 

Reviewer #1: Yes

Reviewer #2: Yes

4. Have the authors made all data underlying the findings in their manuscript fully available?

Reviewer #1: Yes

Reviewer #2: Yes

5. Is the manuscript presented in an intelligible fashion and written in standard English?

Reviewer #1: Yes

Reviewer #2: Yes

6. Review Comments to the Author

Reviewer #1: The authors have thoroughly addressed all the comments and suggestions provided in the previous review round. Their responses are clear, precise, and demonstrate a careful revision of the manuscript. The revised version meets the journal's standards, and I find no further issues or concerns to raise. I recommend the manuscript for publication in its current form.

Reviewer #2: I thank the authors for their engagement with my comments. I am satisfied with the ways in which minor comments were addressed. More substantive points, particularly no.'s 8, 11, 12, 14, 22, and 23 I elaborate on below.

8. This explanation is well received. Thank you for adding it to the Limitations section.

11. Understood, and good point made about introducing an arbitrary classification. I appreciate its addition to the Limitation section.

12. I do not think it is absolutely necessary, as the dates are not so distant, but if possible to procure I think it would be an informative piece of data.

14. Noted. Adding this analysis may be a departure away from what you are working to achieve in this paper, so I do not feel it essential to add. Interesting area for future research!

22. Thank you for adding this. I believe the paper is stronger as a result of this section's addition.

23. The figures are easier to interpret now. I also compliment the authors on the other figures and the addition of Figure 1, very helpful indeed.

Thank you for your efforts. I look forward to seeing this work published.

7. PLOS authors have the option to publish the peer review history of their article (what does this mean?). If published, this will include your full peer review and any attached files.

Reviewer #1: No

Reviewer #2: No

---

## [Author Response · Author response to Decision Letter 2]

7 Feb 2025

We sincerely appreciate the constructive feedback from the editorial team and reviewers. We have carefully revised our manuscript in response to the comments and have addressed each point as outlined below.

Editorial Comments

1. Line 281: Missing reference regarding Australia benefiting from the Regional Expansion of WoS (2006-2008).

o We acknowledge this oversight and have now included a relevant reference to support this statement.

2. Line 278-280: Strong wording about linguistic and systemic bias.

o As suggested, we have rephrased the sentence to align with the editorial team's recommendation:

"This imbalance may reflect a linguistic and systemic bias, where English-speaking countries with well-established research ecosystems are more prominently featured."

3. Line 313-315: Statement about high-impact journals prioritizing certain types of research needs support.

o We have now added a reference to substantiate this claim.

4. Line 254: Discrepancy in the number of OA journals (25,658 in text vs. 24,976 in Figure 2).

o The difference arises because the figure reflects the most up-to-date count after filtering out inactive journals from OpenAlex. We have now updated the text to be consistent with the figure.

5. Line 256: Discrepancy in the number of OA journals indexed in all three databases (4,104 in text vs. 4,094 in Figure 2).

o Similarly, this difference is due to the removal of inactive journals in OpenAlex, which were initially counted in the text but not in the figure. We have now corrected the text for consistency.

Response to Reviewer #1

We deeply appreciate the reviewer’s positive evaluation of our revisions and their recommendation for publication.

Response to Reviewer #2

We are grateful for the reviewer’s thoughtful feedback and have considered all points carefully. Below, we answer each comment:

• Point 8: Thank you for acknowledging our addition to the Limitations section.

• Point 11: We appreciate the reviewer’s recognition of our explanation regarding arbitrary classification.

• Point 12: While the data suggested is not readily available, we acknowledge its potential value and note it as a possible area for future research.

• Point 14: We agree that this additional analysis, while interesting, would shift the focus of our study. We appreciate the reviewer’s understanding on this matter.

• Point 22: We are pleased to hear that the addition of this section has strengthened the paper.

• Point 23: Thank you for the positive feedback on the revised figures. We are glad that Figure 1 has improved the clarity of our results.

We are grateful for the insightful reviews and believe that the revised manuscript is now stronger as a result. Thank you for considering our submission, and we look forward to your feedback.

Sincerely,

The authors.

---

## [Editor Report · Decision Letter 2]

18 Feb 2025

Geographical and Disciplinary Coverage of Open Access Journals: OpenAlex, Scopus and WoS

PONE-D-24-47603R2

Dear Dr. MADDI,

We’re pleased to inform you that your manuscript has been judged scientifically suitable for publication and will be formally accepted for publication once it meets all outstanding technical requirements.

Kind regards,

Alberto Baccini, Ph.D.

Academic Editor

PLOS ONE

Additional Journal Comments:

re: Figure 2 - although you have stated that you have updated the text to match the figure, it appears that the line "striking observation is the presence of 25,658 OA journals exclusively in OpenAlex" is still unchanged from the earlier version, therefore we kindly request that you additionally complete this change prior to submitting the final version of your manuscript, thank you!
---

## [Editor Report · Acceptance letter]

PONE-D-24-47603R2

PLOS ONE

Dear Dr. Maddi,

I'm pleased to inform you that your manuscript has been deemed suitable for publication in PLOS ONE. Congratulations! Your manuscript is now being handed over to our production team.

Kind regards,

on behalf of

Prof. Alberto Baccini

Academic Editor

PLOS ONE